# The Role of Healthcare Providers in Promoting Human Papillomavirus Vaccines among Men Who Have Sex with Men: A Scoping Review

**DOI:** 10.3390/vaccines10060930

**Published:** 2022-06-10

**Authors:** Kok-Yong Chin, Sophia Ogechi Ekeuku, Muhammad Rafie Hamzah

**Affiliations:** 1Department of Pharmacology, Faculty of Medicine, Universiti Kebangsaan Malaysia, Cheras, Kuala Lumpur 56000, Malaysia; 2Kuala Lumpur AIDS Support Services Society, Kuala Lumpur 51000, Malaysia; mdrafie.hamzah@klass.org.my

**Keywords:** Papillomaviridae, perception, physician, sexual minorities, vaccination

## Abstract

Background: The uptake of human papillomavirus vaccines (HPVV) among men who have sex with men (MSM) remains unsatisfactory. Healthcare providers play a crucial role in improving HPVV acceptability and uptake among MSM. This scoping review aims to provide an overview of (1) the perceived role of healthcare providers by MSM, and (2) the knowledge, beliefs and practices of healthcare providers themselves in promoting HPVV uptake. Methods: A literature search was performed with PubMed and Scopus databases using a specific search string. The relevant original research articles on this topic were identified, and the major findings were charted and discussed. Results: The literature search identified 18 studies on the perceived role of healthcare providers by MSM, and 6 studies on the knowledge, beliefs and practices of healthcare providers in promoting HPVV uptake among MSM. Recommendations by healthcare providers and disclosure of sexual orientation were important positive predictors of higher HPVV acceptability and uptake. Sexual healthcare providers were more confident in delivering HPVV to MSM clients compared to primary practitioners. Conclusion: Recommendation from, and disclosure of sexual orientation to healthcare providers are important in promoting HPVV uptake among MSM. The competency of healthcare providers in delivering HPVV to MSM can be improved by having clearer guidelines, education campaigns and better incentives.

## 1. Introduction

Human papillomaviruses (HPVs) are double-stranded DNA viruses that belong to the Papillomaviridae family. Over 200 types of HPV have been identified and they infect the skin and mucosae of the upper aero-digestive and anogenital tract [1]. Of these, the carcinogenic HPV type 16, 18, 31, 33, 35, 39, 45, 51, 52, 56, 58 and 59, and the probably carcinogenic HPV type 68 have been categorised as high-risk type [2]. HPV causes almost all cervical cancer, 90% of anal cancer and 30% of oropharyngeal cancer worldwide [3]. On the other hand, infection of low-risk HPV causes low-grade changes and genital warts on the female cervix, vagina, vulva and anus, as well as the male penis, scrotum and anus [4]. While HPV and its health implications in women are well recognised, the same cannot be said for men [5]. A recent meta-analysis in 2021 reported the global prevalence of anal HPV16 was 13.7% in HIV-negative MSM and 28.5% in HIV-positive MSM, while the prevalence of high-risk HPV was 41.2% in HIV-negative MSM and 74.3% in HIV-positive MSM. These values are higher than HIV-negative (HPV16 1.8%, high-risk HPV 6.9%) and HIV-positive men who have sex with women (HPV16 8.7%, high-risk HPV 26.9%) [6]. These findings showed that men who have sex with men (MSM) and men living with human immunodeficiency virus (HIV) were disproportionally affected by HPV.

HPV vaccines (HPVV) have been developed to prevent infection and HPV-associated diseases. Some of the vaccines available in the market currently include bivalent vaccine (Cervarix, against HPV16 & 18), quadrivalent (Gardasil, against HPV 6, 11, 16 and 18) and nonavalent vaccines (Gardasil 9, against HPV6, 11, 16, 18, 31, 33, 45, 53 and 58) [7]. In the United States, the Advisory Committee on Immunization Practices (ACIP) have recommended routine HPV vaccination for females aged 11 or 12 years since 2006 [8] and males aged 13 through 21 years since 2011 [9]. The ACIP also began recommending catch-up HPVV through the age of 26 years for all populations in 2019 [10]. In the United Kingdom, the Joint Committee on Vaccination and Immunisation (JCVI) has recommended HPVV for females aged 12 or 13 years since 2008, males aged 12 years since 2019 and MSM up to the age of 45 years since 2022 [11]. Recent results from a multinational open-label, long-term extension of a randomised controlled phase 3 trial showed that the quadrivalent HPV vaccines could protect men against anogenital disease related to HPV6, 11, 16 and 18 [12]. Various cross-sectional studies also supported that HPV infection was lower in vaccinated MSM compared to unvaccinated counterparts [13,14,15,16]. A declining trend has been reported for the prevalence of anogenital warts in the United States between 2010 and 2016 among women and men who have sex with men or women, which could be attributed to the effects of HPVV [17].

Despite the efficacy of vaccines, the HPVV uptake among MSM remains unsatisfactory [18]. For instance, among 1651 men living with HIV in Ontario, only 7% received HPVV. Of the unvaccinated, only 40% heard about HPVV [19]. A meta-analysis in 2021 showed that the mean HPVV acceptability rate was 63%, uptake rate was 45% and completion rate was 47% among MSM of 78 studies, mostly performed in the United States [20]. Some of the reasons include the misconception about HPVV being reserved for women, the low perceived threat of HPV-related diseases and the stigma attached to HPVV [21]. A recent systematic review has identified HPVV knowledge and recommendation by healthcare providers are facilitating factors, and high cost and doubts about HPVV effectiveness and safety are barriers to HPVV uptake among the sexual minorities [22].

Improving HPVV uptake requires integrated efforts from multilevel stakeholders [23], especially healthcare providers who act as a point of contact and an important source of information about health issues. They could assist in improving knowledge, recommending vaccines and clearing concerns about HPVV. Thus, the objective of this scoping review was to determine (1) the perceived role of healthcare providers by MSM, and (2) the knowledge, beliefs and practices of healthcare providers themselves in improving vaccine acceptability and uptake among MSM. We hope this review can highlight the importance of healthcare providers in promoting HPVV uptake among MSM and subsequently reducing the morbidity of HPV-related diseases.

## 2. Literature Search

The current review was conducted following the PRISMA guide for scoping review (Appendix A) [24]. A literature search was performed using PubMed and Scopus with the search string (Physicians OR “healthcare providers” OR doctors OR professional OR practitioners) AND (MSM OR “Men having sex with men”) AND (vaccination OR vaccines) AND (HPV OR “Human Papilloma Virus”) in April 2022. The search frame was from the inception of databases until the date of the search. No additional filter was applied in the search. Tracing of references cited in included articles was also performed to ensure relevant articles were included.

All original research articles discussing the role of healthcare providers, not limited to medical doctors, in promoting HPVV acceptance and uptake were included. Both the role of healthcare providers perceived by MSM and the healthcare providers themselves was considered. The subjects of the study were required to involve MSM populations or healthcare providers who engaged with MSM. Articles not written in English, not discussing the role of healthcare providers and not containing sub-analysis for the MSM populations were not included. Reviews, letters, editorials and book chapters that did not include primary data were excluded. Conference proceedings and abstracts were also not included to avoid study duplication.

Endnote X9 (Clarivate, Philadelphia, USA) was used to organise the literature and detect duplication of items. Two authors (K.-Y.C. and M.R.H.) independently screened the titles and abstracts of the articles, and then retrieved the full text for detailed examination based on inclusion and exclusion criteria. Any discrepancies in article inclusion were resolved by discussion with the third author (Ekeuku S.O.). Data extraction was performed by the two authors (K.-Y.C. and M.R.H.). The data extracted include authors (years), study design, characteristics of the subjects, major findings and limitations.

## 3. Results

The literature search yielded 24 results in PubMed and 29 results in Scopus. After removing the duplicates, 35 unique articles were identified and subjected to screening (7 were excluded based on article types, 11 were excluded based on topics). Subsequently, the full text of 17 articles was further evaluated. Two articles were rejected, and seven relevant articles were identified from the reference list of included articles. Ultimately, this review included 24 original research articles for analysis (Figure 1, Appendix A). 

All studies included adopted a cross-sectional observational design. A total of 18 studies investigated the perceived role of healthcare providers in promoting HPVV acceptance and uptake by MSM [25,26,27,28,29,30,31,32,33,34,35,36,37,38], while 6 studies investigated the knowledge, beliefs and practices of healthcare providers themselves in delivering HPVV to MSM [39,40,41,42,43]. The MSM were recruited conveniently from sexual health centres [25,27,31], MSM organisations and venues [28,33,35,37], a university [44], MSM social websites or dating apps [30,34,35,38,45] or Facebook [21,36,38]. Several studies also used national sampling of relevant populations [26,29,32,46]. The healthcare providers surveyed were sexual healthcare workers [39,40,41,47], primary healthcare providers or general practitioners [41,42,47] and staff from community-based HIV/AIDS service organisations [43], recruited conveniently. Only one study used a randomised sampling approach to recruit primary care physicians [42]. All studies were conducted in the western countries, including the United States, United Kingdom, Canada and Australia.

From the point of view of MSM, HPVV recommendation from, and disclosure of sexual orientation to healthcare providers were positive predictors of higher vaccine acceptability [26,28,30,32,33,34,35,36,44,45]. Inversely, the absence of an established relationship with healthcare providers, negative emotions, experience and uneasiness in disclosing sexual orientation to healthcare providers as well as the lack of recommendation from healthcare providers were identified as barriers to obtaining HPVV in qualitative studies [21,35,38,45,46]. A recent visit to healthcare providers and access to healthcare facilities also predict higher HPVV uptake [29,34]. A summary of key findings is presented in Table 1.

Among healthcare providers, those involved specifically in sexual healthcare were more likely to recommend HPVV to MSM, be aware of the existing HPVV recommendation and vaccinate MSM [39,41]. Similarly, staff from community-based HIV/AIDS service organisations were aware of HPVV and willing to encourage their client to talk to their healthcare providers about HPVV and direct them to one of the providers [43]. They also believed that they had a positive role in influencing MSM clients in their decision-making process in taking up HPVV [43]. The primary care physicians and general practitioners were more reserved about vaccinating MSM [41,42]. In the United Kingdom, the general practitioners were less aware of HPVV for young MSM, less agreeable to gender-neutral HPVV and less confident in identifying young MSM who might benefit from HPVV and recommending it to them [41]. In the United States, 70.5% of the primary physicians were aware of HPVV recommendation for MSM but only 13.6% routinely discussed sexual orientation and HPVV with male patients aged 22–26 years [42]. The lack of efficacy data on HPVV on older men and cost were major issues that bothered both sexual healthcare specialists and general practitioners alike in Canada [47]. A summary of key findings is presented in Table 2.

## 4. Discussion

This review explored the role of healthcare providers in promoting HPVV uptake from the perspective of MSM and healthcare providers themselves. From the literature, it was identified that vaccine recommendation by, and sex orientation disclosure to healthcare providers were associated with HPVV acceptability among MSM. A pleasant relationship with and easy access to healthcare providers also promote HPVV uptake in MSM. On the other hand, sexual healthcare providers were found to be more knowledgeable and confident in recommending HPVV to MSM clients, compared to general practitioners and primary physicians. HIV/AIDS outreach workers also believed that they played a positive role in influencing HPVV uptake among MSM.

In general, MSM prioritises healthcare providers’ recommendations in accepting HPVV. For instance, 37.6% of MSM aged < 28 years in France who received physicians’ proposal for HPVV initiated the vaccination compared to 1.9% among those who did not receive the same proposal [38]. Wheldon et al. [35] reported that a doctor’s opinion was the only opinion that matters in influencing MSM’s acceptability of the vaccines. These observations may stem from the lack of knowledge about the HPVV among MSM, despite the high awareness of HPV. Even in studies conducted before HPVV was licensed to be used in men, >70% of MSM surveyed by multiple studies had heard about HPV [25,26,28]. However, only 28–30% of MSM were aware of HPVV [25,27]. In a later online study by Cummings et al. [30], awareness about HPV (87.9%) and HPVV (74.1%) was found to increase. Despite this, uncertainties on HPVV remain. A qualitative study by Wheldon et al. [35] reported that young MSM were uncertain about the side effects and efficacy of HPVV, and doubted that the use of live virus in the vaccine can cause HPV. This is obviously a misconception because HPVVs are based on HPV L1 virus-like particles, which are nanoparticles formed by viral structural proteins but do not have any core genetic material [48]. Thus, HPVV cannot cause an HPV infection. Apart from that, some MSM were also worried about the interaction between HPVV and HIV treatment [33]. All these doubts could be cleared with the assistance of healthcare providers. 

Discussions with healthcare providers would also improve perceived susceptibility and seriousness of HPV, and correct some biases about the vaccines. Many studies report that higher self-perceived susceptibility of MSM to HPV-related disease, and increased severity of the diseases, predicted higher HPVV uptake [26,27,30]. In a study by Moores et al. [31], less than half of the men registered for STI testing and treatment in Ottawa were aware that HPV is the primary cause of anal cancer. In the same study, MSM who discussed anal cancer screening and prevention with healthcare providers displayed higher knowledge about HPVV [31]. 

Recommendation for vaccination would not occur without disclosure of sexual orientation. Multiple studies indicated that disclosure of sexual orientation predicted higher HPVV acceptability and uptake [28,29,30,36]. However, only 38% of young MSM disclosed sexual orientation to their healthcare providers in a survey by Stupiansky et al. [34] among users of an online MSM social networking website. A good relationship with healthcare providers will facilitate the disclosure of sexual orientation. Previous negative experiences with healthcare providers which raised the feeling of shame, awkwardness or being judged prevented disclosure [35]. Some MSM felt the necessity to know the providers’ standpoint on LGBT issues for the fear that they might be biased or incompetent in providing care [35]. This issue is not unique to HPVV. In a survey about communication barriers for HIV and STI preventive services, over half of the adolescent MSM avoided disclosure and discussing sexual health issues with healthcare providers owing to fear of heterosexual bias, exposure of their health information to their parents and beliefs that sexual minorities would not receive equal treatment [49]. 

One of the major challenges is to get MSM, who live in regions where HPV vaccination is not mandatory or pass the age of mandatory vaccination, vaccinated before they are sexually active and exposed to the virus. A meta-analysis has shown early sexual initiation and higher lifetime sexual partners are associated with an increased risk of HPV infection in men [50]. Rank et al. [28] reported median time from sexual debut to first disclosure was 6 years (interquartile range 2–14 years) among MSM in Vancouver. In the same study, 37% of MSM aged < 27 years who never disclosed to any healthcare providers reported >5 sexual partners. In an earlier study, 93% of MSM were willing to disclose their sexual orientation to a provider in exchange for free HPVV, but this finding was only valid until the median age of 20 years (2 years after sexual debut) and after a median of 15 years. The delay in disclosure could nullify the efforts of targeted HPV vaccination among MSM. On the whole, the healthcare providers must create an inclusive and private environment that facilitates disclosure and discussion of sexual behaviours with the client, which will directly contribute to timely vaccine uptake. Alternatively, a gender-neutral school-based HPV vaccination could ensure everyone, including MSM, is protected before exposure. 

In terms of preferred facilities to obtain HPVV, some studies revealed MSM preferred sexual health facilities due to the accepting and non-judgemental environment. However, others indicated that young men might not have access to sexual health facilities and preferred general practitioners [33]. Some also suggested the school nurse as a trusted person to deliver the vaccines [37]. 

It is important to ensure the competency of healthcare providers to educate, identify potential receivers and deliver the vaccines, given the important role they play in HPVV uptake. A survey among UK-based sexual health workers revealed that most of them held positive beliefs on HPVV, including MSM-targeted vaccination (65%), the willingness of MSM to accept HPVV (75%), giving HPVV regardless of age (51%) and providing HPVV through general practitioners and pharmacies (74%). However, only 49% of them believed that they had the skills to identify MSM who would benefit from HPVV and 44% believed that they were sufficiently informed about HPVV for MSM. One study also reported a high level of awareness and willingness to facilitate HPV vaccination for MSM among staff at community-based HIV/AIDS service organisations [43]. These organisations are important in regions with low acceptance of LGBT to promote HPVV, as they tend to be non-judgemental and respect clients’ privacy [51]. 

In stark contrast to sexual healthcare providers, Merriel et al. [41] revealed that 78.95% of general practitioners in the United Kingdom had low to no knowledge of HPVV for young MSM, and were less likely to believe that young MSM would disclose sexual orientation to them, that they did not have the skills to identify who may benefit from the vaccines and were less likely to recommend HPVV to young MSM. In a survey in Florida, 70.5% of the primary care physicians were aware of the HPVV recommendation for MSM but only 13.6% routinely discussed both sexual orientation and HPVV with their male patients aged 22–26 years. As many as 24.5% of the physicians did not discuss either with their male patients [42]. The pessimism among the primary care and general practitioners could hinder the vaccination effort as they might be the first point of contact before the sexual debut of young MSM. An alternative way to circumvent the selection criteria is to make HPV vaccination universal so that everybody would take HPVV regardless of age, gender and sexual orientation.

A qualitative study among self-referred healthcare providers by Nadarzynski et al. [40] has highlighted some of their concerns on HPVV, which include selection criteria (age or history of genital warts), appropriate healthcare setting to offer HPVV (sexual health clinics or general practices) and funding source (central or local). They also cited lack of political and public support, limited access to HPVV by MSM in rural areas, delayed disclosure of sexual orientation to healthcare providers and poor awareness and motivation to complete the vaccination, as barriers to implementing MSM-targeted HPV vaccination. The solutions proposed to overcome these barriers include official guidelines and awareness campaigns. Integrated clinical procedures would also facilitate MSM-targeted vaccination, which include non-judgemental processing in recording sexual behaviours, incentivising the recording of this information, encouraging MSM not attending sexual health clinics to be vaccinated and setting up reminders to complete vaccination [40]. The healthcare providers should also be aware that HPVV provides similar protection against future exposure to HPV regardless of previous exposure. In MSM with a history of anal genital warts, the risk of future genital warts is reduced if they take the HPVV, regardless of age [52]. This point highlights again the importance for everyone to take HPVV regardless of age, previous exposure to HPV, gender and sexual orientation.

There are several common biases in the studies included in this review. These limitations might not be specific for the studies on this topic but all studies involving sensitive populations such as MSM. Firstly, the generalisation of the findings from these studies should be performed with caution. For studies recruiting MSM in healthcare services, they might be more health-conscious than the non-attendees. Studies recruiting subjects in MSM venues are at risk of self-selecting MSM who were comfortable with their sexual orientation. Some studies were performed in regions more open to the LGBT population. Thus, the subjects may not represent the MSM population who are not ready to disclose their orientation or those in rural areas. Some qualitative studies included have a limited sample size and data might not reach saturation due to constraints in recruiting subjects. For the study on healthcare providers, only one study adopted randomised sampling. The others adopted a convenient sampling approach, which might have oversampled healthcare providers with a specific interest in HPVV for MSM. Lastly, all studies were conducted in western countries with wider LGBT acceptance. The results could be very different in countries that marginalise LGBT populations. 

## 5. Conclusions

Healthcare providers play an important role in promoting HPVV acceptability and uptake among MSM. Discussion of sexual orientation and behaviours with MSM clients predicts higher HPVV uptake, but this requires an open and non-judgemental patient–provider relationship. There is a discrepancy between HPVV knowledge, beliefs and practices among sexual healthcare providers and primary or general healthcare providers, which needs to be bridged to optimise HPVV delivery to MSM. Clearer vaccination guidelines, education and incentives will perhaps motivate the healthcare providers to recommend and deliver HPVV to their MSM clients.

## Figures and Tables

**Figure 1 vaccines-10-00930-f001:**
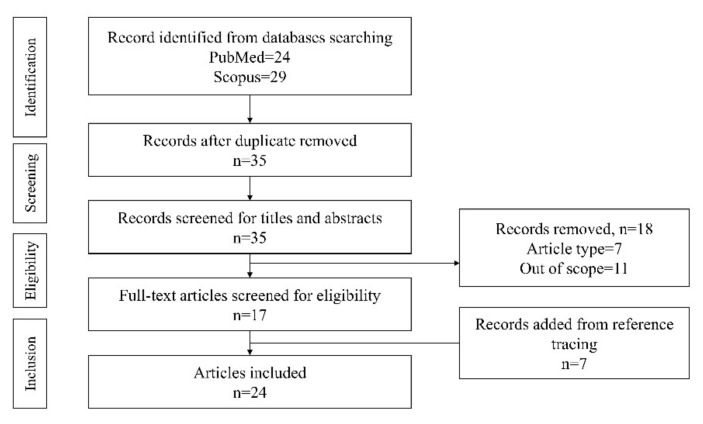
Article selection process.

**Table 1 vaccines-10-00930-t001:** Perceived importance of healthcare providers in promoting HPVV by MSM.

Study	Subjects Characteristics	Major Findings	Notes
Simatherai et al. [25]	MSM attending the Melbourne Sexual Health Centre *n* = 200 Median age = 27 years Age range = 19–71 years	93% would disclose to healthcare professionals they were MSM if they could obtain HPVV for free. This was valid until a median age of 20 years (2 years post-sexual debut) and a median sexual partner number of 15.	Huge challenge to get MSM vaccinated before HPV exposure. Need to address the low level of awareness first. Subjects could be more health-conscious. Suggest vaccination to all boys.
Reiter et al. [26]	A national sample of self-identified gay or bisexual men in the US *n* = 306 Age range = 18–59 years	Higher HPVV acceptability among those who perceived their doctors would recommend it (vs unbeliever, OR 12.87, 95% CI 4.63–35.79) and those who were doubtful (vs unbeliever, OR 3.15, 95% CI 1.47–6.76), ≥5 lifetime-sexual partners (OR 3.39, 95% CI 1.34–8.55) and perceived higher severity of HPV-related disease (OR 1.92, 95% CI 1.18–3.14), perceived higher HPVV effectiveness (OR 1.97, 95% CI 1.27–3.06), perceived higher regrets if they developed an HPV infection if unvaccinated (OR 2.39, 95% CI 1.57–3.61).	Conducted before HPVV is licensed to be used among men. Inclusion of MSM outside the range of recommended vaccination. Willingness might not translate to behaviours.
Colón-López et al. [27]	Men ≥26 years attending an STI Clinic in Puerto Rico *n* = 46	Factors increasing vaccination willingness: —A doctor recommended HPVV (95.7%) —Health insurance reimbursed HPVV (91.3%) —Subjects understand the importance of the vaccine (91.3%) Barriers to be vaccinated: —Low perceived susceptibility towards infection —High cost	Subgroup analysis for MSM was not performed. Very small sample size.
Rank et al. [28]	MSM aged ≥19 years recruited at community venues in Vancouver *n* = 1401	↑ vaccine acceptability was linked with previous diagnosis of genital warts (OR 1.7, 95% CI 1.1–2.6), sexual behaviour disclosure to healthcare providers (OR 1.6, 95% CI 1.1–2.3), annual income ≥ $20,000 (OR 1.5, 95% CI 1.1–2.1), previous hepatitis A or B vaccines (OR 1.4, 95% CI 1.0–2.0) and absence of recreational drug use (OR 1.4, 95% CI 1.0–2.0) Median time from disclosure to first sexual contact: 6.0 years (IQR 2–14 years) 37% of men ≤26 years who never disclosed to any healthcare provider reported ≥6 lifetime sexual partners.	Recruitment at the public, so MSM must be willing to disclose themselves, thus could be more comfortable discussing with healthcare providers. The cost of HPVV was not factored in. Delay in disclosure will increase the likelihood of exposure to HPV infection before vaccination.
Meites et al. [29]	MSM under National HIV Behavioural Surveillance System *n* = 3221 Age range = 18–26	Factors predicting HPVV uptake: —Visiting a healthcare provider last year (aPR 2.3, CI 1.2–4.2) —Ever disclosing male-male sexual attraction/behaviour to a healthcare provider (aPR 2.1, CI 1.3–3.3) —A positive test for HIV infection (aPR 2.2, CI 1.5–3.2) —Any hepatitis vaccination (aPR 2.2, CI 1.5–3.2).	Vaccine status was self-reported. Vaccine completion was not evaluated. Venue-based sampling would exclude those who were discrete about their sexual orientation.
FitzGerald et al. [44]	Subjects were conveniently sampled in a university *n* = 12 Age range = 18–28 years	Healthcare professionals were cited as the main referents to approve subjects receiving HPVV. Subjects mainly referred to general practitioners. Cost, low knowledge level on HPV/HPVV and concerns on side effects were barriers to vaccination.	Small sample size.
Cummings et al. [30]	Email to registered users of the world largest men who seek social or sexual interactions with other men *n* = 1457 Mean age = 22.5 (SD = 2.40) years	↓ HPVV acceptability was linked with HPVV safety concerns (B = −0.262, *p* < 0.01), greater shame associated with HPV infection/disease (B = −0.103, *p* < 0.01), and perceived resistance (B = −0.089, *p* < 0.01). ↑ HPVV acceptability was linked with healthcare provider’s recommendation (B = 0.190, *p* < 0.01), greater worry about HPV infection (B = 0.139, *p* < 0.01), and being tested for an STD in the previous year (B = 0.060, *p* < 0.05). ↑ vaccine uptake was linked with being tested for a sexually transmitted disease in the previous year (OR 3.27, 95% CI 1.87–5.70), disclosure of sexual orientation (OR 2.99, 95% CI, 1.83–4.88), and higher HPV knowledge scores (OR 1.40, 95% CI 1.23–1.59)	Lack of disclosure could lead to no recommendation from doctor to take up HPVV. Low response rate (1457/4801) Self–reported vaccination status.
Moores et al. [31]	Men registered for health services at a sexually transmitted infection testing and treatment clinic in Ottawa, Ontario *n* = 280 Mean age = 37 ± 11.86 years (range 18–69)	16.2% were vaccinated with HPVV. For unvaccinated individuals, only 27.2% talked to healthcare professionals about vaccinations. 74.9% had family doctors and from those, 75% know their clients were MSM. For MSM who had discussed anal cancer screening and prevention with healthcare providers (n = 30), most were knowledgeable about HPVV.	Over-sampling of MSM open about their sexual orientation. Under-sampling of MSM who went to their GP for screening.
Reiter et al. [32]	Harris Interactive LGBT Panel LGBT living in the United States Age range = 18–26 years *n* = 428	13% initiated the vaccination, 54% of them completed all doses. The major reason for vaccination was doctor’s recommendation. 83% of those who received the recommendation for HPVV by a healthcare provider initiated the vaccination. In multivariate analysis, recommendations by healthcare providers remained the strongest correlate of HPVV initiation (OR 110.60; 95% CI 32.67, 374.48).	First study after AICP’s recommendation for routine vaccination of males was released in late 2011. Only includes subjects who self-identified as gay or bisexual. HPV vaccination is self-reported.
Gerend et al. [45]	MSM recruited from a geospatial dating app *n* = 336 Age rage = 18–26	Provider recommendation was the strongest predictor for HPVV uptake (40 times more likely to be vaccinated). Provider’s recommendation was predicted by sexual identity (↑ others vs gay), ethnicity (↓ Hispanic vs White), condomless anal sex (↑ yes vs no) and HIV status (↑ yes vs no). Lack of recommendation, lack of HPV/HPVV knowledge, not disclosing sexual identity, low susceptibility for HPV and concerns about vaccine safety are barriers to vaccination.	Self-reported vaccination and recommendation. Exclusion of those under 18 years.
Nadarzynski et al. [33] *	MSM from community-based LGBTQ venues and organisations *n* = 33 Median age = 25 years (IQR: 21–27), Age range=16-60 years	All MSM would accept HPVV if offered by a healthcare professional. Barriers: accessing healthcare services or discussing same-sex experience with healthcare professionals, efficacy of vaccines, side effects, fear of needles, fear of interaction between HPVV and HIV treatment. The majority preferred sexual health clinics as a means to reach out to MSM due to openness, some preferred GP because young men had limited access to sexual health clinics.	Self-selection bias. Education level not determined. The area surveyed was more open to the LGBTQ population. The absolute incidence of anal cancer not disclosed to prevent subjects underestimating the risk, which might change the attitudes towards vaccines.
Wheldon et al. [35]	*n* = 9 from student pride groups *n* = 13 from sexual networking application used by MSM Interview in person = 14 Telephone = 8 Age range = 18–26 years, mean = 22 years	Interpersonal influence of HPVV acceptance: doctor’s opinion was the most important, some stated influence of more senior gay friends. Family support was mixed due to alienation. External control factors: out-of-pocket cost, uncertainties about where to get vaccinated, lack of established relationship with providers, convenience (distance, schedule). Self-efficacy to ask for HPVV was mixed due to uneasiness to disclose sexual orientation to healthcare providers. Relationship with provider: sometimes negative, impacting disclosure and interactions. Some feel ashamed, awkward and judged. Felt the need to understand healthcare providers’ standpoint on LGBT issues for fear that the provider might be biased or incompetent in providing care.	Small sample size. Specific geographical area. All well-educated with healthcare insurance.
Stupiansky et al. [34]	US users of an online MSM social and sexual networking website *n* = 1751 Mean age = 22.7 years	38% disclosed same-sex relationship to their healthcare provider. Increased ≥1 dose of HPVV was linked with: —Higher disclosure to friends/family —Recent sexually transmitted disease history —Visiting a healthcare provider in the past year —Searching for sexual health information online and disclosure to healthcare providers were important mediators in the relationship between these predictors and vaccine uptake Having visited a healthcare provider in the past year was the most important predictor of disclosure of MSM behaviour.	The high dependence on disclosure reflects HPV vaccination is especially dependent on practice of individual providers. Users of social and sexual networking websites are high-risk groups. The Black population is underrepresented.
Nadarzynski et al. [36]	MSM recruited via advertisement via Facebook *n* = 1508 Median age = 22 years Age range = 14–63 years	89% would accept HPVV if a healthcare provider offered it. HPVV acceptability was positively associated with: —access to sexual health clinics [OR 1.82, 95% CI 1.29–2.89] —disclosure of sexual orientation to a healthcare provider [OR 2.02, CI 1.39–3.14] —positive HIV status [OR 1.96, CI 1.09–3.53] After receiving HPVV information, the acceptability was positively associated with: —↑ perceived HPV risk (OR 1.31, CI 1.05–1.63) —↑ perceived severity of HPV infection (OR 1.89, CI 1.16–3.01) —↑ perceived HPVV benefits (OR 1.61, CI 1.14–3.01) —↑ perceived HPVV effectiveness (OR 1.54, CI 1.14–2.08) —↓ perceived barriers to HPV vaccination (OR = 4.46, CI 2.95–6.73)	Convenience sampling method. Targeting only men who were already comfortable disclosing their sexuality online. Recall bias and social desirability.
Kesten et al. [37] **	MSM recruited from LGBTQ organisations, university information days, university student union	65% had never discussed HPVV with a healthcare provider. Mean age of participants willing to disclose sexuality to healthcare providers = 18.3 years (range: 11–23 years). The most comfortable setting to receive HPV vaccine was LGBTQ-specific services than genitourinary medicine clinics. Thematic analysis: A good relationship with general practitioners or sexual healthcare providers is important for HPVV acceptance. The school nurse was suggested as a trusted person to deliver the vaccine.	Small sample size. Self-selection—participants may be more comfortable with their sexuality.
Gerend et al. [21] *	Men identified as gay, bisexual or queer recruited via Facebook or a local LGBTQ health and development program *n* = 29 Mean age = 22.66 (SD = 2.30) Age range = 18–26 years	Some were not sure HPVV is effective for sexually active men. Doubts on the number and timing of doses, age, side effects of HPVV. Provider played a central role in subjects’ decision to be vaccinated. Some providers seemed uncomfortable asking subjects’ sexuality. Some felt stigmatised or judged. Some were hesitant in asking for HPVV if they had to disclose sexuality. The level of comfort relied on their relationship with the providers.	Small sample size. Subjects are from regions with higher socioeconomic status.
Petit and Epaulard [38]	MSM under the age of 27 recruited via Facebook, community website or dating application *n* = 2094	Among 1728 with a family physician, 9.9% was proposed HPVV (9.1% for those ≤ 27 years), 60.6% disclosed sexual orientation. 17.9% ≤ 27 years had received the vaccine. 37.6% received the proposal accepted HPVV, compared to 1.9% among those who did not.	Self-selection of subjects with greater interest in sexual health. Might have a higher vaccination rate than the general MSM in France.
Jaiswal et al. [46] *	Sexual minority men recruited from a larger cohort study of emerging sexual minority adults in New York City. *n* = 38 Mean age = 25.82 (SD = 0.95) years Age range = 24-27 years	Healthcare providers did not explain the importance of HPVV adequately. Healthcare system did not follow up with clients to complete vaccination.	No in-depth exploration of the topic. Not readily generalisable to sexual minorities of other areas.

Abbreviation: ↑, increased; ↓ decreased; ACIP, Advisory Committee on Immunization Practices; aPR, adjusted prevalence ratio; B, regression coefficient; CI, confidence interval; GP, general practitioner; HPV, human papillomavirus; HPVV, HPV vaccine; LGBTQ, lesbian, gay, bisexual, transgender, queer; MSM, men who have sex with men; aPR, SD, standard deviation; OR, odds ratio. Notes: All studies included in this table are cross-sectional quantitative type, apart from those marked with * and **, which are qualitative and mixed quantitative/qualitative type.

**Table 2 vaccines-10-00930-t002:** Knowledge, beliefs and practices of healthcare providers in providing HPVV.

Study	Subjects Characteristics	Major findings	Notes
Nadarzynski et al. [39]	UK-based sexual health workers (i.e., consultants, nurses, health advisors) *n* = 325 (70% female, 46% doctors, 75% in sexual health clinics)	65% recommended targeting MSM for HPVV. 3% believed that HPV poses little cancer risk in MSM to make vaccination necessary. 75% believed that the majority of MSM would want to receive HPVV. 60% believed that HPVV would promote MSM to engage with sexual health services. 3% believed that HPVV increased the likelihood of unsafe sex among MSM. 26% believed in individual assessment of MSM attending sexual health clinic. 74% believed HPVV should be offered by GPs or pharmacies. 51% believed all HPVV offering to MSM should not be based on age. 17% believed it is too late to vaccinate if MSM are sexually active. 49% believed they have the skills to identify MSM that would benefit from HPVV. 44% believed that they are sufficiently informed about HPVV for MSM. Sexual healthcare providers who were vaccinating men had less odds to disagree that MSM are not at risk of HPV-related cancers and that MSM-targeted HPV vaccination is worthwhile (OR 0.34, 95 CI% 0.20–0.70). They also believed they had higher knowledge levels about issues related to HPVV and MSM (OR 8.49, 95% CI 4.50–15.1). Nurses were more likely to agree with individual assessment in MSM-targeted HPV vaccination (OR 3.32, 95% CI 1.69–5.65).	Risk of self-selection. The response rate cannot be determined. Not include GPs and pharmacists.
Nadarzynski et al. [40] *	UK-based self-referred healthcare providers (13 doctors, 3 nurses, 3 health advisers) involved in sexual healthcare *n* = 19	Issues: healthcare providers were not sure about selection criteria (younger/without a history of genital warts), appropriate healthcare setting (sexual health clinics/GP) and source of vaccination funding (central/local). Barriers: Lack of political and public support (become a sex vaccine), limited access to HPV vaccination by MSM (rural area), delayed disclosure of sexual orientation to healthcare providers, identification of eligibility, poor awareness and motivation to complete vaccination. Facilitating factors to increase coverage: official guidelines, awareness campaigns and integrated clinic procedures (non-judgmental processing in recording sexual behaviours, incentivise recording, encouraging HPVV for MSM not attending sexual health clinics, reminders to complete vaccination).	Gender-neutral vaccination is preferred over MSM-targeted screening. Effects will be compromised if MSM are not willing to attend sexual health clinics or disclose sexual orientation. Suggested effective use of social media such as Facebook, poster advertising and text messages. Small sample size. Self-selection bias from healthcare providers with a particular interest in HPV vaccination.
Merriel et al. [41]	General practitioners and sexual healthcare providers (including genitourinary medicine consultants, doctors-in-training and nurses working in sexual health clinics) *n* = 87 (38 GPs and 49 sexual healthcare providers) Mean age = 40.71 years with a median 14 years of experience (IQR 8. 24).	Sexual healthcare providers were more likely to vaccinate a young MSM, and aware of the recommendation (adjusted OR 0.03, 95% CI 0.01, 0.11) and perceived self-sufficient to engage in informed discussion with HPVV (adjusted OR 0.04, 95% CI 0.01, 0.14). 78.95% general practitioners indicated no to low knowledge of HPV vaccination for young MSM, compared to 12.24% among sexual healthcare providers. GPs were less likely to agree on sex-neutral (adjusted OR 0.30, 95% CI 0.09, 0.98) or HPV vaccination MSM (adjusted OR 0.30, 95% CI 0.09, 0.98), or young MSM would want to be vaccinated (adjusted OR 0.13, 95% CI 0.04, 0.41). GPs were less likely to believe that a young person would disclose their sexual orientation (adjusted OR 0.17, 95% CI 0.06, 0.50), less confident that they could identify young MSM who may benefit from HPVV (adjusted OR 0.03, 95% CI 0.01, 0.15), and in recommending HPVV to young MSM (adjusted OR 0.04, 95% CI 0.01, 0.18). Barriers to deliver HPVV to young MSM: GP—65.79% no time, sexual healthcare providers—staff availability. Solution: GP—73.68% additional training, sexual healthcare providers—51.43% computer prompts.	Convenience sampling approach. Pre-determined survey statements did not allow reason to be given for the opinion. Small sample size.
Wheldon et al. [42]	Primary care physicians in Florida *n* = 770 drawn from American Medical Association Physician Masterfile	70.5% knew HPVV recommendation for MSM. 13.6% routinely discussed both sexual orientation and HPVV with male patients aged 22-26 years (high potential group). 24.5% did not discuss either. HPVV discussion was positively associated with awareness of the physicians on the recommendation for MSM (OR 3.49; 95%CI 1.80–6.74). Physicians with low HPV knowledge were more likely to discuss sexual orientation and HPVV. Postulation: Those with high knowledge levels failed to act on HPVV recommendations based on assessment data.	Suggest the use of electronic medical systems to prompt providers regarding specific recommendations. Non-probability-based sampling. Modest response rate. Not examining communication in real time 51% response rate.
Wigfall et al. [43]	Staff from three community-based HIV/AIDS service organizations *n* = 30 Mean age = 47.7 (SD = 12.5) years	100% were aware of HPV and 77% were aware of HPVV. 67% were aware that HPV causes anal cancer. 91–95% were willing to prompt MSM and female clients to talk to a healthcare provider about HPVV. 86–95% were willing to direct clients to adult safety net HPVV providers. 59–67% thought they could exert a positive influence on MSM and female client’s HPVV decision-making. 63% thought HPV stigma was a barrier to HPV cancer prevention tool.	Small sample size—provider level participant. Gay stigma as a potential healthcare access barrier was not evaluated.
Grace et al. [47] *	13 physiciansand 2 clinical researchers in Canada. Most affiliated with HPV-SAVE project. 7 were HIV/sexually transmitted disease specialists 6 general practitioners	The subjects were in favour of HPVV and were not concerned with its safety. They would recommend HPV to MSM < 27 years, those with health insurance, or HIV-positive patients regardless of age and insurance. HPVV recommendation for older men with HIV, the lack of evidence of benefits of vaccinating MSM > 26 years could affect the recommendation. In these situations, the recommendation was based on patients’ contact with HPV and their sexual history. Discussion on HPV and HPVV was the priority. Both healthcare providers and patients had initiated the discussion. Cost was a major factor inhibiting discussion.	The subjects showed a high degree of knowledge on the HPV research and recommendation practice than other physicians.

Abbreviation: CI, confidence interval; GP, general practitioner; HPV, human papillomavirus; HPVV, HPV vaccine; IQR, interquartile range; MSM, men who have sex with men; SD, standard deviation; OR, odds ratio. Notes: All studies included in this table are cross-sectional quantitative type, apart from those marked with *, which are the cross-sectional qualitative type.

## Data Availability

Not applicable.

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
