# Peer review of "The Role of Healthcare Providers in Promoting Human Papillomavirus Vaccines among Men Who Have Sex with Men: A Scoping Review"

_vaccines, 2022, doi:10.3390/vaccines10060930_

Round 1
Reviewer 1 Report
Chin et al. have performed a literature review of the role of healthcare providers promoting HPV-vaccine to men who have sex with men (MSM). They used a specific search in PubMed and Scopus. In addition, they identified relevant articles from the reference list of the included articles. In conclusion, healthcare providers play an important role in promoting HPV vaccine among MSM. Discussion of sexual orientation and behaviors with MSM predicts higher HPVV uptake, but this requires an open and non-judgemental patient provider relationship. There is a discrepancy between HPVV knowledge, beliefs and practices among sexual healthcare providers and primary or general healthcare providers, which needs to be bridged to optimize delivery of HPV vaccine to MSM. Clearer vaccination guidelines, education and incentives will perhaps motivate the healthcare providers to recommend and deliver HPV vaccine to their MSM clients.
The claims are properly placed in the context of the previous literature. The experimental data support the claims. The manuscript is written clearly enough that it is understandable to non-specialists. The authors have provided adequate proof for their claims, without overselling them. The authors have treated the previous literature fairly. The paper offers enough details of methodology so that the experiments could be reproduced.
Minor revisions
Line 181-183, discussion, “A qualitative study by Wheldon et al. [35] reported that young MSM were uncertain about the side effects and efficacy of HPVV, and doubted that the use of live virus in the vaccine can cause HPV”
Please reformulate. There is no live virus in the HPV-vaccine. The HPV-vaccines are based on HPV L1 virus-like particle (VLP). VLPs are virus-resembling nanoparticles formed by viral structural proteins but do not have any core genetic material. The HPV-vaccine cannot cause an HPV infection.
Line 208-209, discussion, “One of the major challenges is to get MSM vaccinated before they are sexually active and exposed to the virus”
I disagree. There is no problem to vaccinate MSM before they are sexually active and exposed to the virus. The solution is very simple: gender-neutral school-based HPV-vaccination. Usually, school-based HPV-vaccination is for 12–13-year-old children, but the vaccine is approved for boys and girls from 9 years old, and to be sure to vaccinate the children before getting sexually active, the vaccination program can start at 9 years old boys and girls. If you vaccinate all 9-year-old boys and girls, you have also vaccinated all the MSM before getting sexually active and exposed to the virus.
Line 238-242, discussion, “In stark contrast to sexual healthcare providers, Merriel et al. [41] revealed that 78.95% of general practitioners in the UK had low to no knowledge of HPVV for young MSM, and were less likely to believe that young MSM would disclose sexual orientation to them, that they don’t have the skills to identify who may benefit from the vaccines, and are less likely to recommend HPVV to young MSM”
The solution is to make it simple. There are no selection criteria for who may benefit from the vaccine. Everybody should take the HPV-vaccine regardless of age, gender and sexual orientation.
Line 249-251, discussion, “A qualitative study among self-referred healthcare providers by Nadarzynski et al. [40] has highlighted some of their concerns on HPVV, which include selection criteria (age or history of genital warts)”
Is a history of genital warts an argument for or against HPV-vaccination? Many people believe that the efficacy of the HPV-vaccine is dependent of previous exposure to HPV. This is wrong. The HPV-vaccine gives the same protection against future exposure of HPV regardless of previous exposure. In MSM with a history of anal genital warts, the risk of future genital warts is reduced if they take the HPV-vaccine, regardless of age (Swedish 2014). Everybody should take the HPV-vaccine regardless of age, previous exposure for HPV, gender and sexual orientation.
References
Swedish KA, Goldstone SE. Prevention of anal condyloma with quadrivalent human papillomavirus vaccination of older men who have sex with men. PLoS One. 2014 Apr 8;9(4):e93393. doi: 10.1371/journal.pone.0093393. PMID: 24714693; PMCID: PMC3979673.
https://pubmed.ncbi.nlm.nih.gov/24714693/
Author Response
Dear reviewer,
Thank you for reviewing our manuscript. We are grateful for your constructive comments, and they are responded in the attached response sheet. All changes in the manuscript are highlighted in yellow.

Reviewer 2 Report
This is a very interesting scoping review trying to shed light on two sides of the same “coin” 1) the role of healthcare providers as perceived by MSM, and 2) the knowledge, beliefs, and practices of healthcare providers themselves in improving vaccine acceptability and uptake among MSM. Removing barriers in both of these two aspects would definitely improve vaccination rates in MSM. The manuscript is well written and designed. Found only minor issues, mainly readability related.
A few comments:
There is probably a typo in figure 1, removed records should be 18 instead of 16?
Table 1 could be (in my opinion preferred to gain some space (and clutter) by removing the second column (Study design) and marking the single study that is not “Cross-sectional quantitative study” and then modifying the caption. This may improve readability. Similarly table 2 (all studies have a similar design). Moreover, if the tables are in landscape form would be easier to read, in the current format it was rather difficult…
There are minor linguistic issues, for example, “A summary of key findings was presented in Table 2” perhaps use present tense?
Author Response
Dear reviewer,
Thank you for reviewing our manuscript. We are grateful for your constructive comments, and they are responded in the attached response sheet. All changes in the manuscript are highlighted in yellow.

This manuscript is a resubmission of an earlier submission. The following is a list of the peer review reports and author responses from that submission.